# Data-driven electron-diffraction approach reveals local short-range ordering in CrCoNi with ordering effects

Haw-Wen Hsiao[1,2,6], Rui Feng[3,6], Haoyang Ni [1,2], Ke An [3], Jonathan D. Poplawsky [4], Peter K. Liaw [5] & Jian-Min Zuo [1,2] ✉

The exceptional mechanical strength of medium/high-entropy alloys has been attributed to hardening in random solid solutions. Here, we evidence non-random chemical mixing in a CrCoNi alloy, resulting from short-range ordering. A data-mining approach of electron nanodiffraction enabled the study, which is assisted by neutron scattering, atom probe tomography, and diffraction simulation using first-principles theory models. Two samples, one homogenized and one heat-treated, are observed. In both samples, results reveal two types of short-range-order inside nanoclusters that minimize the Cr–Cr nearest neighbors ($L1_2$) or segregate Cr on alternating close-packed planes ($L1_1$). The $L1_1$ is predominant in the homogenized sample, while the $L1_2$ formation is promoted by heat-treatment, with the latter being accompanied by a dramatic change in dislocation-slip behavior. These findings uncover short-range order and the resulted chemical heterogeneities behind the mechanical strength in CrCoNi, providing general opportunities for atomistic-structure study in concentrated alloys for the design of strong and ductile materials.

High-entropy alloys (HEAs) belong to a new class of materials that are chemically concentrated in less-explored phase spaces. Since their initial discovery[1,2], HEAs have attracted tremendous interest for their remarkable structural diversity[3–6] and unique electrical, magnetic, and mechanical properties[7,8], including high strength and great ductility[9]. While HEAs were initially modeled as random solid solutions[10], recent works have focused on the possibility of chemical short-range ordering (CSRO) and its unusual effects on microstructural, mechanical, and electronic properties[11–14]. Particular attention has been given to the medium-entropy alloy (MEA) CrCoNi. This alloy, which belongs to a family of Cr-Mn-Fe-Co-Ni alloys[2], has demonstrated superior mechanical properties[15], irradiation resistance[15,16], and quantum criticality at low temperatures[17]. Especially, the close proximity of ferromagnetic

(Co and Ni) and antiferromagnetic (Cr) interactions can lead to magnetic frustration, and the concept of magnetic-driven CSRO has inspired multiple experimental[18–20] and theoretical[21–23] works.

The standard analysis of CSRO is through single-crystal diffraction[24]. CSRO produces characteristic diffuse scattering, and when measured by X-ray and neutron diffraction, a quantitative determination of atomic pair-correlations can be made[24–26]. However, the single-crystal diffuse scattering analysis of MEA/HEAs has been performed rarely, and the interpretation of diffuse scattering from samples with different thermomechanical histories is extremely challenging[13]. Electrons can work with polycrystals, but the application is hampered by the lack of strong elemental contrast in HEAs, which makes direct imaging of CSRO at the atomic resolution a challenging

[1]Department of Materials Science and Engineering, University of Illinois Urbana-Champaign, 1304W Green St, Urbana, IL 61801, USA. [2]Fredrick Seitz Materials Research Laboratory, University of Illinois Urbana-Champaign, 104S Goodwin Ave, Urbana, IL 61801, USA. [3]Neutron Scattering Division, Oak Ridge National Laboratory, Oak Ridge, TN 37831, USA. [4]Center for Nanophase Materials Sciences, Oak Ridge National Laboratory, Oak Ridge, TN 37831, USA. [5]Department of Materials Science and Engineering, The University of Tennessee Knoxville, Knoxville, TN 37996, USA. [6]These authors contributed equally: Haw-Wen Hsiao, Rui Feng. ✉e-mail: jianzuo@illinois.edu

task. For the face-centered-cubic (fcc) CrCoNi, Zhang et al.[19] reported electron diffuse streaks along {111} in the energy-filtered [110] zone-axis diffraction patterns (DPs) in a heat-treated sample, while Zhou et al. reported the $(\bar{3}11)/2$ diffuse spot in the [112] zone axis diffraction pattern in a cold-rolled sample[20]. However, the short-range order (SRO) parameters were not determined, and it is far from being clear whether these observations support the theory predicted CSRO[12,21,23]. In the absence of strong experimental evidence, it has been questioned whether CSRO in CrCoNi is a negligible effect[27].

Here we develop a data-driven electron-diffraction approach for the determination of local CSRO, using scanning electron nanodiffraction (SEND) with energy filtering (EF), which overcomes the limitations of the traditional diffuse-scattering analysis with orders of magnitude improvement in the spatial resolution and electron-scattering power. The diffraction analysis is complemented by small-angle neutron scattering (SANS) and atom probe tomography (APT) analyses of chemical fluctuations at a sub-nm scale, and supported by atomic-resolution electron imaging and diffraction simulations using first-principles theory models[23].

## Results

### Ordering in CrCoNi

We prepared two CrCoNi samples from the same ingot, as described in "Methods". Following the initial homogenization at 1200 °C for 48 h for both samples, one was water-quenched (Sample WQ), while the other was heated treated (Sample HT) at 1000 °C for an additional 120 h aging, followed by furnace cooling. Both samples show the same fcc structure ($a = 3.565$ Å for Sample WQ and $a = 3.564$ Å for Sample HT) and similar mechanical response (Fig. 1a, b). The diffuse streaks along {111} directions reported by Zhang et al.[19] are observed in both Samples WQ and HT here by EF-SAED (energy-filtered selected-area electron diffraction) (Fig. 1c, e). Atomic-resolution Z-contrast images obtained by scanning transmission electron microscopy (STEM) show relatively uniform contrast for both samples with no obvious evidence of CSRO (see Fig. 1d, f). This finding is consistent with the previous report by Sales et al.[7].

The seemingly sameness of the two CrCoNi samples from the above results demonstrates a major conundrum in the MEA/HEA research, namely, the lack of sensitive probes among current characterization methods for the chemical subtlety in these materials[3]. To overcome this challenge, we developed a nanodiffraction data-mining approach for the detection of CSRO (Fig. 2, with details in Suppl. Note 1). Briefly, a nanometer-sized convergent beam (1.1 mrad) is first formed for coherent nanodiffraction. Second, an energy filter removes the inelastic-scattering background for diffraction pattern (DP) recording. Third, a stack of DPs is collected in a two-dimensional (2D) scan for a hyperspectral, four-dimensional (4D), dataset using a pixelated detector (Fig. 2a). Data mining of the 4D dataset then allows for diffraction imaging. This powerful approach, known as 4D-STEM, has been demonstrated for the mapping of electrical, magnetic, and strain fields[28–30]. The data-mining procedure in Fig. 2b–d extends 4D-STEM to diffuse-scattering analysis. The key to the analysis is the cepstral STEM-based imaging of nanoclusters (NCs) giving strong diffuse scattering[31]. Their detection then allows for an identification of diffuse patterns from these NCs. The DP grouping, template identification and refinement are subsequently used to determine the specific types of diffuse patterns and their distribution. The sensitivity of detecting CSRO is improved as the amount of electron exposure is multiplied by the number of scan points, $N \sim 10^4$, while the spatial resolution is simultaneously improved from hundreds of nm in EF-SAED to 1 nm in SEND.

Figure 3a shows three types (A, B, and C) of diffuse-scattering patterns that were identified along the [011] zone axis from $10^4$ DPs collected from the two samples. The Bragg diffraction peaks and the spatially-invariant diffuse background in these patterns were removed, using the method described by Shao et al.[31] (Suppl. Note 1). Among the three patterns, Types A and B are two variants of the same type with diffuse-intensity peaks at the special points of {111}/2, while in Type C, the broad diffuse peaks are observed at the special points of (100) and (110). Diffuse scattering at these special points were previously observed in CuPt and CuPd, with the $L1_1$- and $L1_2$-type CSRO, respectively[32]. Notably, the $L1_2$-type diffuse peaks are split, as marked by the arrow in Fig. 3c, and the same feature is also present in CuPd[32].

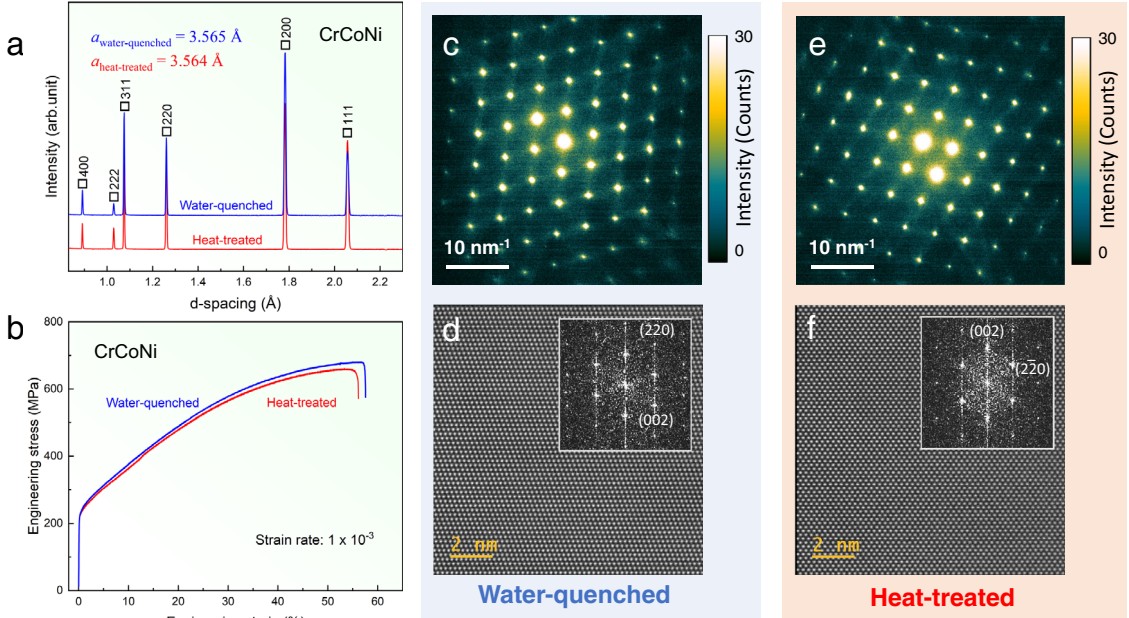

**Fig. 1 | Characterization of the CrCoNi samples prepared by water-quench and heat-treatment. a** Neutron-diffraction patterns, showing the presence of the single fcc phase in both samples. **b** The measured tensile stress-strain up to fracture at the strain rate of $1 \times 10^{-3}$ s$^{-1}$. **c, e** Energy-filtered electron-diffraction patterns and **d, f** atomic-resolution STEM images. The insets in **d, f** are power spectra of the images. The diffraction patterns in **c, e** are displayed in the color scale for the intensities between 0 and 30, in digital counts.

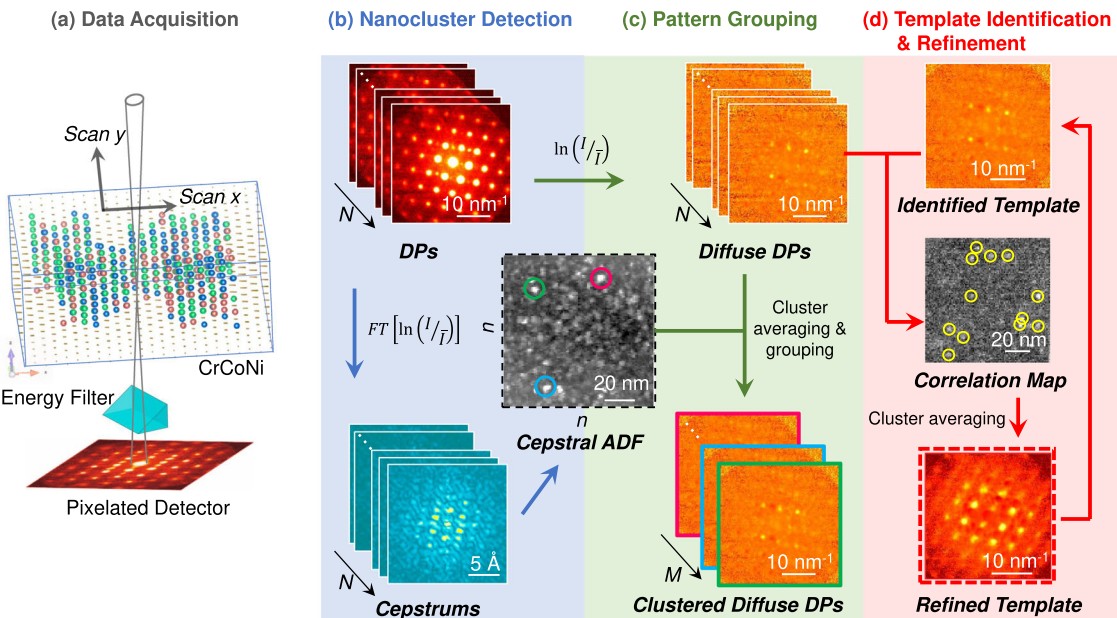

**Fig. 2 | Data mining of diffraction patterns (DPs) for CSRO detection in CrCoNi.** Four steps (**a**–**d**) are involved. **a** Data acquisition, energy-filtered diffraction patterns (DPs) in a scan are recorded using an ~1 nm sized electron probe, with a pixelated detector. **b** The DPs are transformed into difference cepstra, which are then used to form cepstral annular dark-field (ADF) images for nanocluster (NC) detection. **c** Bragg reflections are removed, using the log method, and diffuse DPs belonging to NCs are averaged and grouped together. **d** Diffuse DP templates are identified and used to detect the CSRO-strengthened NCs by the correlation method. The diffuse DPs from the same CSRO-strengthened NCs are then averaged, and the process is iterated to yield the refined templates. $N$ is the number of collected DPs ($10^4$), $M$ is the number of distinct cluster diffuse DPs ($10^1$), and $n$ ($n^2 = N$) is the number of scan points. ADF stands for annular dark field imaging, using the cepstral signals here.

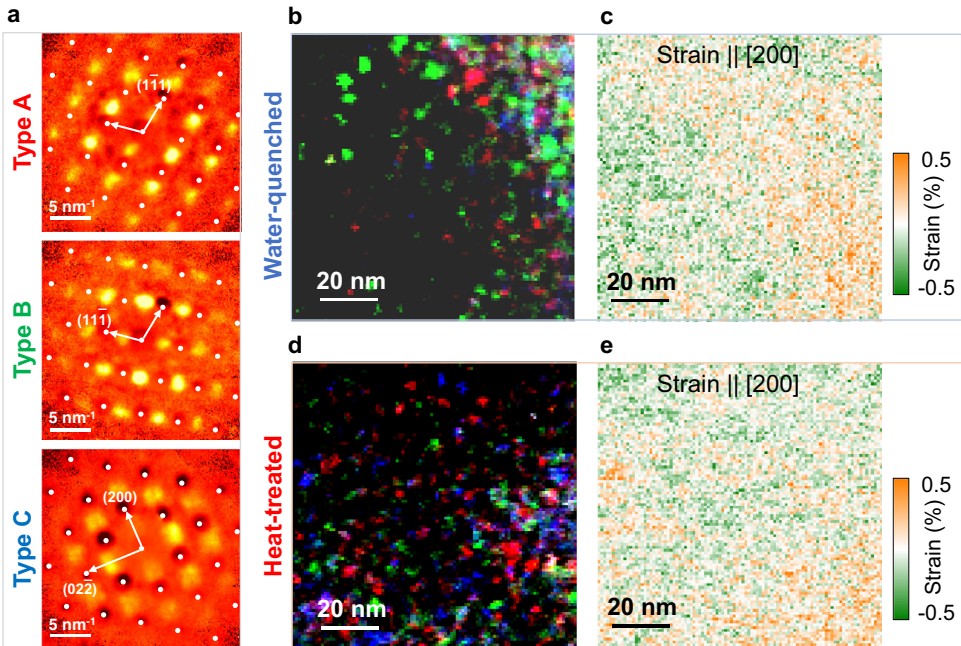

**Fig. 3 | The identified CSRO diffuse-scattering patterns and their distribution in CrCoNi. a** Three types (A, B, and C) of diffuse-scattering patterns along [011], detected by EF-SEND in both water-quenched (WQ) and heat-treated (HT) samples (the white dots mark Bragg peak positions). **b** and **d** are composite color images formed by integrating the diffuse-scattering intensity for Type A (red), Type-B (green), and Type-C (blue) for the WQ and HT samples, respectively. **c** and **e** show the measured strains along the [200] direction from Bragg peaks recorded by EF-SEND for the two samples, respectively. The strong diffuse scattering is found in nanoclusters and their dominance changes from being Types A and B in Sample WQ to being Type C in Sample HT.

Figure 3b, d show two color composites of three images obtained by integrating the Type A, B, and C diffuse intensities in the hyperspectral dataset, respectively, for Samples WQ and HT. Both composites show that the observed diffuse scattering came from local NCs. In Sample WQ, the NCs are dominantly $L1_1$-type (red and green), while in Sample HT, the $L1_1$-type NCs are smaller (red and green), while $L1_2$-type NCs (blue) become more dominant. The distribution of the CSRO-strengthened NCs also changes, from being heterogeneous in Sample

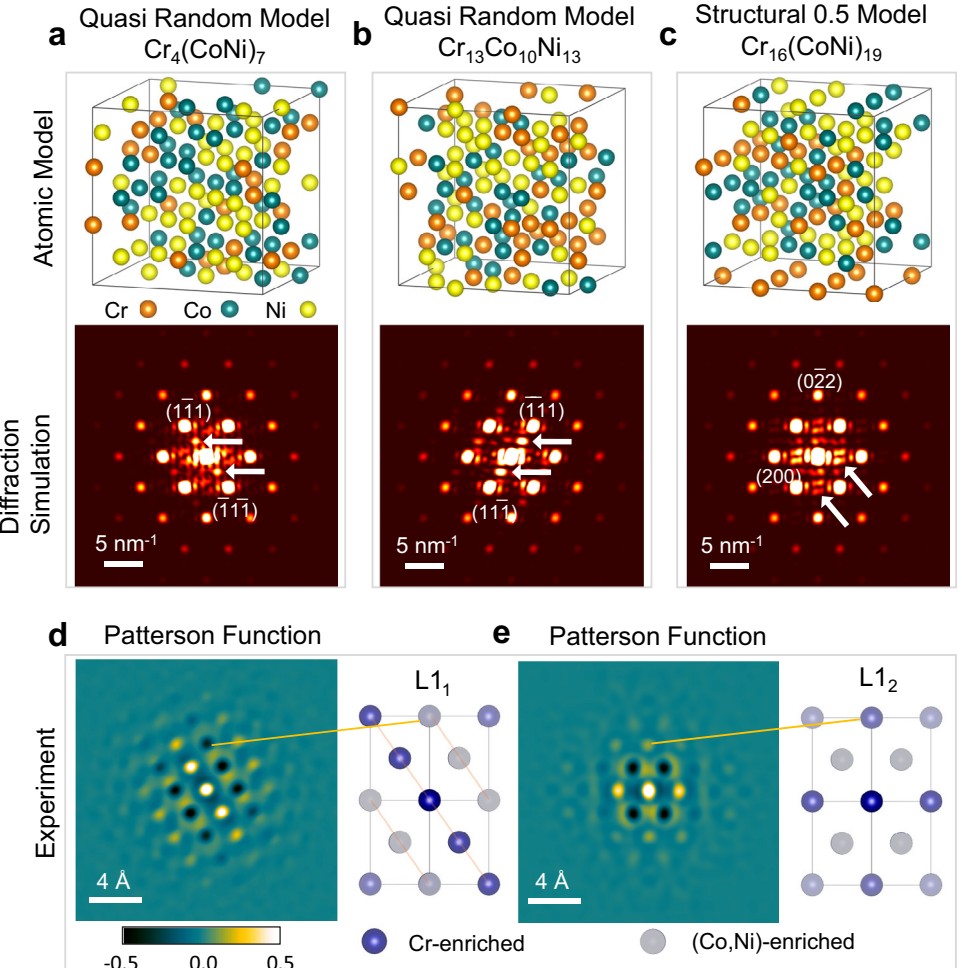

**Fig. 4 | Short-range ordering in CrCoNi.** Atomic structural models from first-principles calculations with (**a**) and **b** from quasi-random configurations and **c** from CSRO configuration (Structural 0.5 model)[23] and their corresponding simulated diffraction patterns. **d** and **e** are experimental two-dimensional Patterson functions and their structural model. The models shown here are selected from an ensemble with the CSRO composition varies from being stoichiometric to off-stoichiometric, which can be poor or rich in Cr (for details, see Suppl. Note 3).

WQ to comparatively homogeneous in Sample HT. This difference is also seen in the strain maps obtained from the Bragg peak analysis[33] of the same hyperspectral datasets (Fig. 3c, e and Suppl. Note 2). The observed strain homogenization correlates with the reduction of $L1_1$-type NCs, which suggests that these NCs are likely lattice distorted. This observation is consistent with the first-principles prediction of tetragonal-like lattice distortion in the CSRO models[23].

**The atomic structure models of CSRO**

CSRO in CrCoNi has been examined theoretically by Tamm et al.[21] and Ding et al.[12], using the Monte-Carlo method combined with density functional theory (DFT) calculations. Their calculations were summarized in the Warren-Cowley SRO parameters ($\alpha_{ij}$, with $i$ and $j$ for each element)[24]. The negative $\alpha_{CrNi}$ and $\alpha_{CrCo}$ and positive $\alpha_{CrCr}$ were predicted for a preference of additional Cr–Co and Cr–Ni nearest neighbors at the expense of the Cr–Cr pairs. The origin of CSRO in CrCoNi was suggested as magnetic[21–23]. The study of CrCoNiFe shows that the anti-ferromagnetic Cr is surrounded by the ferromagnetic Ni, Fe, and Co in an alloyed $L1_2$ structure driven by a reduction in energy[21,22].

To collate with theoretical predictions, we examined the library of atomic-structure models built by Walsh et al.[23] through diffraction simulations (Suppl. Note 3). These models were built with different degrees of CSRO and spin ordering, covering a range of compositions[23]. Figure 4a, b, c present three models selected from a

subset that yield diffuse peaks at the special Brillouin zone points. The models in Fig. 4a, b belong to the simulated quasi-random configurations with $\alpha_{ij} = 0$, while the model in Fig. 4c is one of the so-called "structural 0.5" models, as simulated with $\alpha_{CrCr} = 0.5$ and $\alpha_{CrCo} = \alpha_{CrNi} = -0.25$[23]. The diffuse intensity tends to be stronger in the off-stoichiometric models, i.e., the models in Fig. 4a, b, c all deviate from the stoichiometric composition of CrCoNi. They are part of the trends that we found with the $L1_1$-type features occurring most frequently in the off-stoichiometry quasi-random models and the $L1_2$-type most frequently observed in the off-stoichiometry CSRO models.

Next, we examine the atomistic origin of the $L1_1$- and $L1_2$-type diffuse scattering, using Patterson functions obtained from diffuse patterns in Fig. 3a. The negative nearest-neighbor Patterson peaks in Fig. 4e indicate the $L1_2$-type ordering. In contrast, Patterson-function peaks in the same {111} plane have the same sign in the $L1_1$-type ordering, while the signs are opposite between neighboring planes (Fig. 4d). For an atomic site, $i$, next to a centered atom at 0, the Patterson-function peak, $P_{0i}$, in a binary alloy relates to the SRO parameter $\alpha_{0i}$, according to $P_{0i} \propto m_0 m_i \alpha_{0i}$, with $m$ stands for the elemental composition[24]. In a ternary alloy, $P_{0i}$ is weighted by the difference in the atomic-scattering potential, and for CrCoNi, the largest difference is between Cr and (Co, Ni) over the experimental range of scattering angles (Suppl. Fig. 1). Based on this trend, we conclude that the negative $P_{01}$ for the nearest neighbors is associated with the

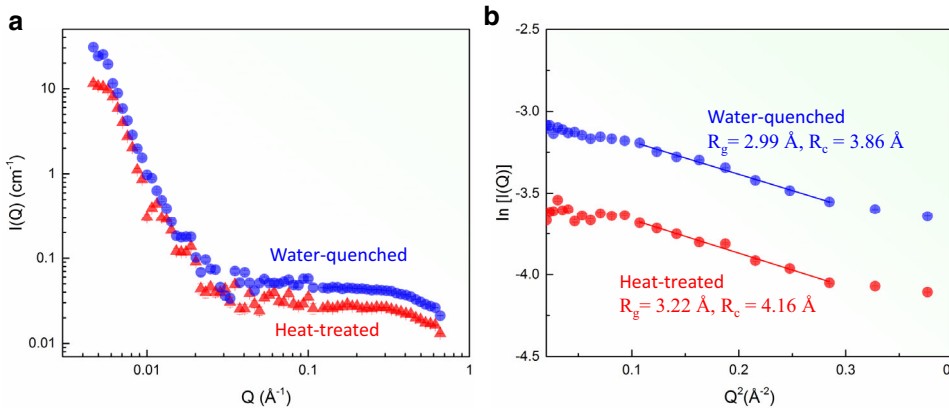

**Fig. 5 | Local chemical clustering detected by small-angle neutron scattering (SANS). a** SANS-intensity distribution as a function of momentum transfer, Q, of the water-quenched and heat-treated CrCoNi samples. **b** Guinier plots plotting ln[I(Q)] vs. Q² and Guinier fit at a high Q range.

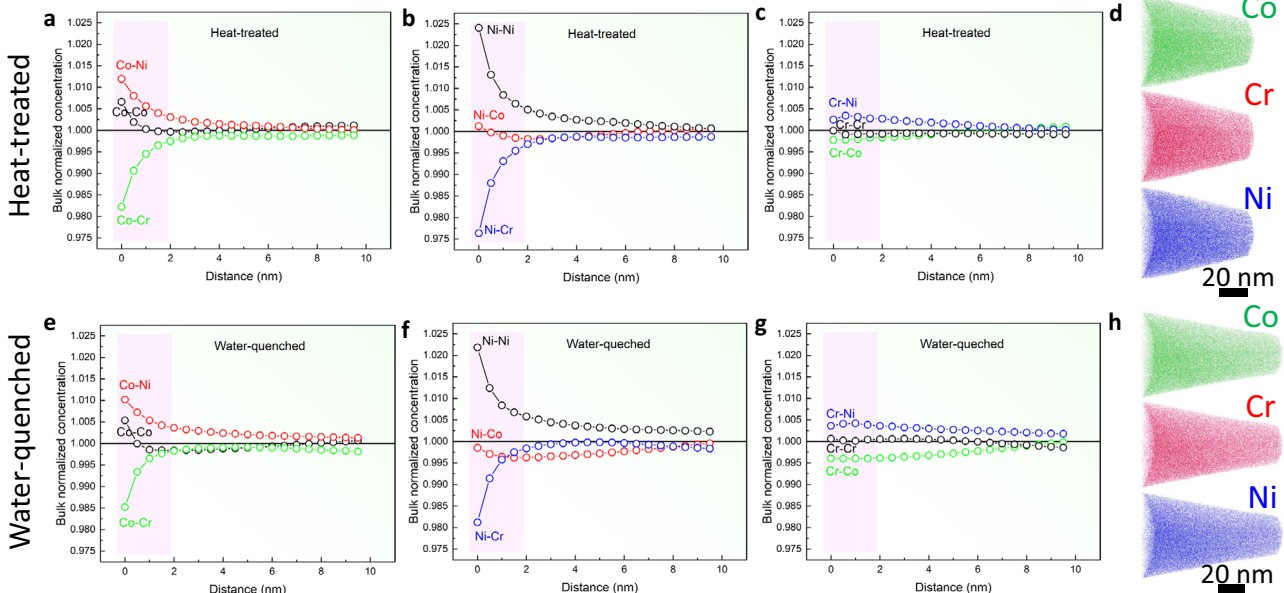

**Fig. 6 | APT partial radial distribution functions (RDFs) of the heat-treated and water-quenched CoCrNi samples. a**, **b**, **c** and **e**, **f**, **g** are the calculated partial RDFs from the APT data with Co, Ni, and Cr as the center atom, respectively, for heat-treated and water-quenched CoCrNi samples. Note that the partial RDFs were normalized by the averaged composition for the APT-analyzed volume. **d** and **h** are APT atom maps of the constituent elements (Co, Cr, and Ni) for the two samples, respectively.

formation of unlike pairs of Cr-Co and Cr-Ni in the L1₂-type ordering. The estimated $\alpha_{CrCo}$ and $\alpha_{CrNi}$ are at −0.18 for the L1₂ CSRO, which is consistent with the Monte-Carlo DFT theory predictions[12,21] (details in Suppl. Note 4). The L1₁-type ordering, on the other hand, goes beyond the nearest neighbors to the distance of ~4.4 Å, and it is formed by Cr and (Co,Ni) clustering on alternating {111} planes.

CSRO can be accompanied by local chemical clustering, leading to chemical fluctuations in the sample. One indicator of such fluctuations is the large misfit-volume in CrCoNi[23,27]. However, the quantification of local chemical fluctuations is difficult. We performed chemical analyses on the CrCoNi samples, using multiple probes. First, STEM energy-dispersive X-ray (EDX) chemical mapping at 1-nm spatial resolution shows no evidence of decomposition (Suppl. Note 5). Second, SANS detects additional scattering objects within the solid-solution matrix with the Guinier radius of the gyration, $R_g$ ~ 3.0 Å, in both Samples WQ and HT (Fig. 5 and Suppl. Note 6). Third, the reconstructed APT volumes of two samples also show no signs of decomposition (Fig. 6 and Suppl. Note 7). A closer inspection of the APT data,

based on the partial radial distribution function (RDF) analysis (Suppl. Note 7) clearly shows the evidence for Ni clustering extending to few nm in radius and negative Ni-Cr and Co-Cr correlations. The latter indicates Cr depletion in Ni- or Co-rich clusters, while the Ni-Co correlation in Ni-rich clusters is slightly negative (Fig. 6b, e). Compared to Ni, the measured Cr-Cr correlation is close to 1, with the effects of Cr-poor and rich regions largely canceled out.

### Impacts of CSRO on dislocation slips

Experimentally, we find that the promotion of the L1₂-type CSRO can have a pronounced impact on dislocation slip. Figure 7 summarizes our TEM observations along two crystal orientations. In Sample WQ, wavy dislocations were seen to be mixed with localized planar slips, which change drastically to extensive planar slips in Sample HT. The planar slip was also observed by Zhang et al.[19]. In Sample WQ, localized planar slips sometimes form dislocation multipoles (MPs) (Fig. 7a and Suppl. Figs. 11, 12). These features can be attributed to the hardening effects of the finely dispersed L1₁-type CSRO-strengthened NCs[34]. In

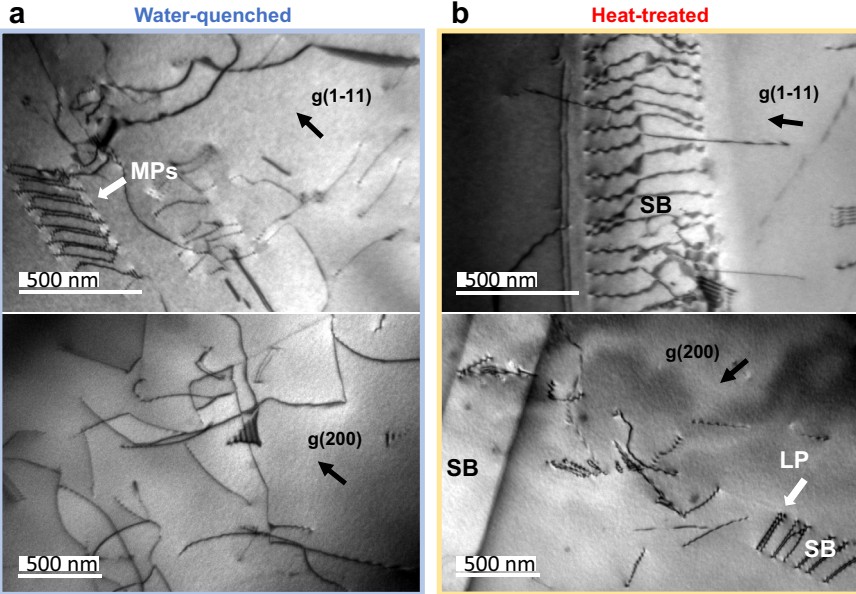

**Fig. 7 | Alteration of dislocation-slip behaviors by short-range ordering.** Chemical short-range ordering is observed in both samples with the L1$_2$ type dominantly found in Sample HT. **a** Two-beam bright-field images of Sample WQ shows long and curved dislocations. Multi-poles (MPs) are also observed here. **b** Dislocations in Sample HT form extensive slip bands (SBs), where paired dislocations (PDs) are commonly seen in the SBs. The scale bars are 500 nm in length.

Sample HT, the leading dislocations in the planar slip tend to form paired dislocations (PDs) at a short spacing, compared to other dislocations in the slip band (Fig. 7b). The formation of PDs is due to the destruction of the L1$_2$-type CSRO by the leading dislocation, which creates higher energy at the diffuse antiphase boundary. The second dislocation partially heals the diffuse antiphase boundary created by the lead dislocation and thus, reduces the energy cost for dislocation motion[35,36]. The destruction of CSRO introduces the so-called glide plane softening effect[34] for pronounced planar slips, as seen in Fig. 7b.

## Discussion

The discovery of the theory-predicted L1$_2$-type CSRO supports the significant role of magnetic interactions in CrCoNi. While the importance of magnetism in the development of alloy microstructures and deformation mechanisms has been known for many decades[37], the atomistic mechanisms have only been addressed recently, largely through computations[12,21,23,38]. The splitting in the diffuse peaks from L1$_2$ CSRO (Fig. 3a), which is consistent with what Ohshima and Watanabe observed in Cu-Pd, can be related to the Fermi surface (FS) shape[26,32], through the so-called FS nesting mechanism[39]. Its observation in CrCoNi suggests that in addition to magnetic interactions, the Fermi surface instability also drives CSRO.

At the sub-nm scale, CSRO minimizes Cr–Cr nearest neighbors (L1$_2$) or segregate Cr on alternating close-packed planes (L1$_1$). While these two types of CSRO can be formed without altering the chemical composition, diffraction modeling suggests that the CSRO-strengthened NCs are likely off-stoichiometry. For example, the diffuse peaks at {111}/2 positions (Fig. 3a), which characterize the L1$_1$-type ordering, are reproduced by diffraction simulations in the off-stoichiometric, quasi-random, models constructed by Walsh et al.[23]. The configurations in these models are either Cr or Co/Ni poor, and their formation is likely associated with the elemental diffusion driven by magnetic interactions.

At the nm-length scale, the APT partial RDF analysis detects Ni-rich and Cr-poor NCs of a few nm in size (Fig. 6 and Suppl. Note 7). The volume-averaged elemental correlations are small, at ~2% or less above or below that of random solutions. The small correlations can be explained by the presence of both elemental-poor and rich NCs in a randomly mixed matrix. The alloy composition is largely uniform at the tens of nm-length scale (Suppl. Fig. 10). The NCs of several nm seen in Sample WQ by diffraction are areas with similar diffraction patterns. The same diffraction features can be produced by the nm-sized elemental-rich or poor volumes, according to our diffraction modeling results. Thus, these large NCs likely contain smaller clusters, whose composition fluctuates.

The yield strengths of the two differently prepared samples are similar at 227 and 224 MPa for Samples WQ and HT, respectively (Fig. 1). The measured yield strengths are in between the values reported by Zhang et al.[19] (205 and 255 MPa, for the homogenized and heat-treated samples, respectively). Inoue et al.[40] also reported the similar mechanical properties in their differently-treated samples. We attribute the similar yield strength of our samples to the presence of CSRO in both. The presence of CSRO in the homogenized sample was not detected by electron imaging, previously[19]. Our as-recorded HAADF STEM images also show no obvious difference between the two samples (Fig. 1). To further investigate these issues, we performed the cross-correlation analysis on atomic-resolution high-angle annular dark-filed (HAADF) STEM images of Sample WQ using template matching (Suppl. Note 8). A spatially averaged single-atomic-column image is used as the template. The resulted cross-correlation coefficient image measures sensitively to the degree of atomic-column deviations. The result reveals both L1$_1$- and L1$_2$-type CSRO in regions of a few nm in sizes (Fig. 8). This observation supports our diffraction analysis and also the recent discovery of L1$_2$ CSRO in the homogenized sample by Inoue et al., using lattice-resolved APT[40]. The observation of CSRO in the homogenized samples shows that CSRO forms rapidly during cooling, similar to Au-Ag and Au-Pd[41]. The rapid formation is possibly assisted by the quenched-in atomic vacancies, which has been shown to have a large effect on the CSRO formation kinetics[41].

Lastly, we note that neither the L1$_1$ nor L1$_2$ CSRO models produce the $(\bar{3}11)/2$ diffuse spot observed by Zhou et al.[20] in their cold-rolled CrCoNi plate sample. This trend, together with our observations of CSRO-strengthened nanoclusters in the matrix of randomly mixed solid solutions, demonstrates experimentally the multiplicity of local bonding preferences in CrCoNi, dependent on the thermal and mechanical history. These discoveries are made possible with the establishment of a sensitive diffraction characterization method, which overcomes the

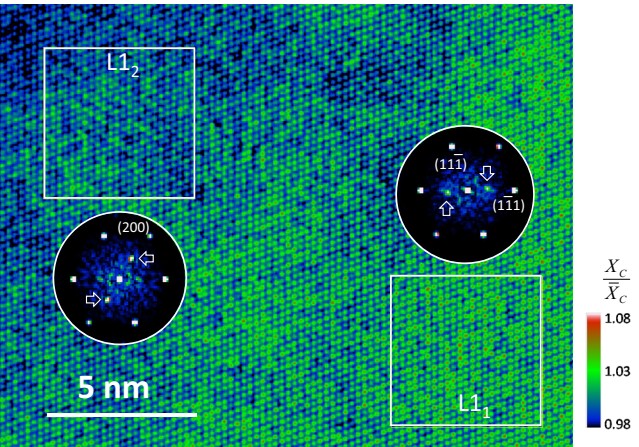

**Fig. 8 | Evidence of short-range ordering in Sample WQ from atomic resolution imaging.** The image is obtained by template matching of the spatially averaged atomic column with the recorded HAADF-STEM image, using the cross-correlation procedure described in Suppl. Note 8. The L1$_1$ and L1$_2$ SRO are detected in two boxed regions, as evidenced by their power spectra. The cross-correlation image is normalized with the mean equals to 1. The color bar indicates the range of the normalized cross-correlation coefficients ($X_C$).

limitations of previous approaches based on single-crystal diffraction[42] or electron diffraction of selected volumes[19,20,32].

The direct measurement and analysis of electron diffuse scattering here unambiguously settles the prior question of whether CSRO exists in this alloy. The observation of diffuse peaks at or near special points inside the Brillouin zone indicates a commonality with CSRO in binary alloys from the contribution of valence electrons, but with a special twist of strong magnetic interactions in CrCoNi. The significance of these confluent factors is highlighted by the dramatical alternation of dislocation-slip behavior with the promotion of the L1$_2$ CSRO. Thus, short-range ordering in CrCoNi driven by magnetic and electronic instabilities leads to ordering effects, including nm-scale chemical inhomogeneities and ordering-dependent dislocation slip behaviors. The methodology advancement here overcomes a major characterization challenge in search of strong and ductile multi-principal elements alloys.

## Methods

### Sample preparation and materials characterization
The CrCoNi alloys were prepared from high purity (>99.9% weight percent) Cr, Co, and Ni by arc melting and then drop casting under an argon atmosphere. To ensure the compositional homogeneity, the melting and casting processes were repeated five times. The prepared alloys were then cut into two pieces and homogenized at 1200 °C for 48 h. After that, one was immediately cooled by water quenching, and another was aged at 1000 °C for 120 h, followed by furnace cooling to room temperature. To avoid the oxidation effect, both samples were sealed in a vacuumed quartz capsule. The differently prepared samples were then fabricated into tensile bars for the tensile-mechanical testing.

The monotonic tension was performed on an MTS Model 810 servohydrauli machine at room temperature with a strain rate of $1 \times 10^{-3}\,\text{s}^{-1}$. The gauge section of the tensile samples has a dimension of $25.4 \times 3 \times 2$ mm. Sample surfaces were mechanically polished to 1,200 grits before the tests. The tensile strains were measured by an MTS extensometer. Three tensile specimens were tested to ensure the reliability of the results.

Neutron diffraction was performed on the VULCAN Engineering Materials Diffractometer at the Spallation Neutron Source (SNS), Oak Ridge National Laboratory (ORNL)[43,44]. Small-angle neutron scattering (SANS) was collected on the CG2 General-Purpose SANS at the High

Flux Isotope Reactor (HFIR)[45]. Two-sample detector distances of 12 m and 1.5 m were used to measure the low Q range of 0.0061 to 0.117 Å$^{-1}$ and the high Q range of 0.044 to 0.722 Å$^{-1}$, respectively.

The APT specimens were fabricated using the standard lift-out and sharpening methods, as described by Thompson et al.[46]. Wedges were lifted out, mounted on Si microtip array posts, sharpened using a 30 kV Ga$^+$ ion beam, and cleaned employing a 2 kV Ga$^+$ ion beam. The APT experiment was run utilizing a CAMECA LEAP 4000XHR in voltage mode with a 50 K base temperature, 20% pulse fraction, a 0.5% detection rate, and a pulse repetition rate allowing for all elements to be detected. The APT results were reconstructed and analyzed using the CAMECA's interactive visualization and analysis software (IVAS 3.8).

### Transmission electron microscopy
The TEM samples were prepared from the end of the tested-tensile bars by mechanical cutting and polishing. The samples were further thinned down to electron transparency by twin-jet chemical etching, using an electrolyte consisting of 95% (volume percent) ethanol and 5% perchloric acid at a temperature of −40 °C and an applied voltage of 30 V. TEM and STEM were carried out, employing a JEOL 2010 TEM (JEOL USA, operated at 200 kV) and a Themis Z STEM (Thermo Fisher Scientific, USA, probe corrected and operated at 300 kV), respectively. Atomic-resolution images were collected, using a focused probe with a semi-convergence angle of 21.4 mrad and the detector inner cutoff angle of 40 mrad. For dislocation analysis, the samples were tilted to different two-beam diffraction conditions. The X-ray EDS area analysis was performed on thin sample areas of ~50 nm in thickness (identified by the convergent beam electron diffraction or CBED), using the Themis Z STEM.

### Energy-filtered scanning-electron nanodiffraction
Regions close to where the EDS analysis was conducted were selected for diffraction analyses. EF-SEND was performed, utilizing the Themis Z STEM in the microprobe mode with a beam current of 40 pA, 1.1 mrad semi-convergence angle, and 1.2 nm in full-width half-maximum. A drift correction algorithm was applied to reduce the sample-drift effect. The DP datasets were collected in a $100 \times 100$ pixel-scan over sample areas of $100 \times 100$ nm$^2$ and 40 to 50 nm in sample thickness. Energy filtering was achieved, employing a Gatan image filter (GIF) camera and the energy selection slit width of 10 eV. Supplemental Fig. 2 provides examples of the record DPs.

### Data availability
The hyperspectral electron diffraction datasets, electron diffuse scattering templates, and STEM images generated in this study have been deposited in the University of Illinois databank at https://doi.org/10.13012/B2IDB-4432073_V1. Other data that support the analysis here are available from the corresponding author (Jian-Min Zuo, jianzuo@illinois.edu) upon request.

### Code availability
The strain analysis tool, imtoolbox, can be downloaded from github at https://github.com/flysteven/imToolBox. Cepstral and Patterson function analysis code are available at https://doi.org/10.13012/B2IDB-4432073_V1. Other code that supports the analysis here are available from the corresponding author (Jian-Min Zuo, jianzuo@illinois.edu) upon request.

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

## Acknowledgements

H.W.H. and J.M.Z. are supported by Intel, a SRI grant from GCOE UIUC and NSF DMR-1828671. R.F. thanks for the support from the Materials and Engineering Initiative at the Spallation Neutron Source (SNS), Oak Ridge National Laboratory (ORNL). P.K.L. is supported by the National Science Foundation (DMR-1611180 and 1809640) and the US Army Research Office (W911NF-13–1-0438 and W911NF-19–2-0049). A portion of this research used resources at the High Flux Isotope Reactor (HFIR) and SNS, a U.S. Department of Energy (DOE) Office of Science User Facility operated by the ORNL. APT research was supported by the Center for Nanophase Materials Sciences (CNMS), which is a US Department of Energy, Office of Science User Facility at Oak Ridge National Laboratory. The authors would like to thank Dr. Ken Littrell and Dr. Dunji Yu for their help in neutron-scattering measurements. The authors also thank James

Burns for his assistance in performing APT sample prep and experiments. This manuscript has been authored by UT-Battelle, LLC under Contract No. DE-AC05-00OR22725 with the U.S. Department of Energy. The United States Government retains and the publisher, by accepting the article for publication, acknowledges that the United States Government retains a non-exclusive, paid-up, irrevocable, world-wide license to publish or reproduce the published form of this manuscript, or allow others to do so, for United States Government purposes. The Department of Energy will provide public access to these results of federally sponsored research in accordance with the DOE Public Access Plan (http://energy.gov/downloads/doe-public-access-plan).

## Author contributions

H.W.H., R.F., and H.N. contributed to the experimental data collection, analysis, and their description. K.A. provided the support and guidance on the neutron scattering. J.D.P. performed the APT experiments and analyzed the data. P.K.L. provided the HEA sample, and comments and discussions on the paper. J.M.Z. directed the research, developed the data-mining approach, contributed to data analysis, and wrote the manuscript with help of all authors.

## Competing interests

The authors declare no competing interests.
