## [Peer Review File · Nature Communications]

Title: ~~Short Range Orders of L11 and L12 Type Revealed in the CrCoNi Medium Entropy Alloy with Novel Ordering Effects~~REVIEWER COMMENTS

Reviewer #1 (Remarks to the Author):

This manuscript presents an important and well-motivated novel methodology for analyzing chemical short range order.

The method is simple (in a good way) and well-motivated, and is backed by important materials science conclusions and complementary characterization that form the main results of the paper.

I have some comments related to specification and reproducibility of the methodology that I think are important to address before publication, particularly because the manuscript presents novel methodology.

* data and source code availability (and statement?)

I strongly encourage you to publish full supporting datasets and software along with this manuscript. This is the norm in the materials data science community, and is critical for the transparency and reproducibility of applied data science work, and can significantly increase your impact in the materials data science community by more effectively enabling others to adopt and extend your methodology.

* Clarity of the methodology

I think revising Figure 1 somewhat to more clearly illustrate *how* the results are obtained would be very helpful for this manuscript, since some basic understanding of how the template-matching algorithm works will be useful for the context of this manuscript, which really does highlight the novel methodology. The current figure 1 is hard to understand at this level without fully reading and understanding the supplementary materials.

I think this could be clarified by revising a few of the panels and improving the figure caption. Alternatively, step 4 could be its own figure.

I would suggest:

The panel titled "Nanocluster Detection": show all detections using the same color since this step comes before template identification and assignment

The panel titled "Nanocluster Diffuse Patterns" could be more clearly named "Identify pattern templates" or similar

Finally, the panel at bottom right could show a schematic image with detections of each class, annotated with the colors corresponding to the template patterns shown in the top right panel.

I would recommend using the image at bottom right with 10 yellow nanocluster detections for both the detection and classification/assignment panels, changing the colors of the detections in the assignment

panel.

I think some revisions along these lines would go a long way towards clarifying the method.

* Full documentation of methodology

The data mining approach is not discussed in detail in the manuscript, and is not fully specified in the supplementary materials. Since the materials science results are enabled by this novel methodology, it is important to discuss in detail the specific details of each component of the analysis pipeline.

** peak detection algorithm

document which algorithm is used, and any important hyperparameter settings, as well as a brief discussion of the criteria / data used to select.

** template generation and matching algorithm

#+BEGIN_QUOTE

First, three diffraction templates belonging to CSRO are created from the cluster averaged log difference DPs

#+END_QUOTE

how does this work? manual identification of templates from a list of nanocluster average log-difference diffraction patterns? Random selection? Initialization from k-means clustering? Lots of interesting possibilities

The number of templates to include seems to be an important hyperparameter. How was the value of 3 selected and validated? Strong prior information based on a physical hypothesis is an acceptable justification, but warrants an explicit argument.

Next, you template-match individual diffraction patterns with cross-correlation against the template patterns. What is the value of the cross-correlation threshold, and how was it selected and validated?

Patterns attens selected through this procedure are averaged to refine the templates, in an iterative procedure? It's unclear from the manuscript whether templates are refine once or multiple times (until what criterion is met?)

The connection to k-means clustering is not clear, and I think it would be worth discussing in a bit more detail which aspects are similar. Certainly there are similarities in the cluster/template assignment step, and the iterative updating/refinement of the templates. Important differences to discuss seem to be 1. initialization strategy and 2. your method does not assign every instance to a template, in contrast to k-means which does.

**** oxide filtering**

It's not explicitly discussed in the methodology -- how are oxide nanoclusters filtered out before performing template generation and matching? Are nanocluster detections manually flagged by inspecting diffraction patterns? That is implied by the discussion of indexing NiO reflections. Is there an automated procedure used to match and remove oxide nanoclusters before the template generation and matching process?

Reviewer #2 (Remarks to the Author):

This manuscript reports two types of nanocluster short-range-orders (SRO) in an important single phase FCC medium entropy alloy CrCoNi. The unique properties of this alloy has been proposed to originate from SROs but the type of ordering structures has not been identified previously. The work is similar to the Nature work from Minor's group in 2020. Both works tackle the SRO issue with state of the art TEM tools and then show that SRO can be modified by processing techniques, and can affect mechanical behaviors. The conditions of samples are also exactly the same. Despite the fancy new techniques, the manuscript is not very well-organized. In particular, key information regarding the nanoclusters are confusing and self-conflicting in the present manuscript. The work may have new implications to the SRO in CrCoNi, but the authors must address the following issues before this work can be further considered for publication.

1. The authors claim that whether the existence of SRO has not been confirmed in previous literature. Is it really so? Even in Minor's work? Or is the existence of SRO already confirmed but the structure remains unknown? Please clearly explain.
2. In general, one would expect the alloys to behave more like a random solid solution at higher temperature. This is exactly what Minor's work observed. Here, however, the NCs were larger, and the local strain was more uneven in the WQ sample, which suggests the opposite. The authors should explain and discuss this in detail. They should also discuss how the degree of SRO change with temperature in the alloy.
3. Why were the NCs not observable by HR or HRSTEM in the present work? In particular, both AP and HRTEM/HRSTEM should be able to image at least the L11 NCs, which were 6 nm in size. Such images could provide direct evidence for the existence of NCs and could also provide useful information of the SRO structure.
4. Could you provide a rough idea of the thickness of the TEM foil?
5. Information regarding the composition of the NCs were confusing. On the one hand, the PDF suggest Cr depletion in the NCs for both samples. The atomic structural models in Fig. 3 are also based on off-stoichiometric compositions – but towards the opposite direction (Cr-rich side). On the other hand, the

L12 NCs were described as “segregate Cr on alternating close-packed planes” – which can be achieved without any composition change in the NCs. The authors should explain these discrepancies and describe the composition of the L11 and L12 NCs. Also, if there is Cr depletion in the NC, there should be corresponding Cr-rich regions in the alloy, was those observed?

6. Please provide Cr-centered PDF.

7. The authors claim that for L12 “segregate Cr on alternating close-packed planes”, but it seems to me that the NC with Cr on alternating close-packed planes in Fig. 3d is L11. It there a mistake?

8. Quality of the manuscript is not quite pleasing. For example, the authors say that “In Sample WQ, localized planar slips sometimes form dislocation multipoles (MPs) (Fig. 4a and Suppl. Note 8)”. But there is no Note 8 in the supplement. There are also quite a few typos, for example, “Pattern functions” in the caption of Fig. 3 is clearly wrong.

Reviewer #3 (Remarks to the Author):

The paper by Hsiao presents a novel data-mining approach of electron-nanodiffraction patterns, complemented by other experimental and simulation techniques, to investigate short-range ordering (SRO) in the Medium entropy CrCoNi alloy (MEA). The results reveal two types of SRO that can dramatically the dislocation-slip behavior and hence the mechanical response of the alloy. This work illustrates convincingly the relevance of this new approach to study SRO in MEA/HEAs alloys. The presence of two types of SRO is clearly demonstrated and the experimental observations support the theory-predicted CSRO. In addition the writing is clear and the results are well presented. For these reasons, I am happy for the work to be published in Nature Communications, once the comments in the document attached are considered.

In what follows, we address the reviewers' comments regarding our submission. For reference, the reviewers' original comments are quoted in *italic*.

Reviewer #1 (Remarks to the Author):

Comment 1-0: *This manuscript presents an important and well-motivated novel methodology for analyzing chemical short range order. The method is simple (in a good way) and well-motivated, and is backed by important materials science conclusions and complementary characterization that form the main results of the paper.*

I have some comments related to specification and reproducibility of the methodology that I think are important to address before publication, particularly because the manuscript presents novel methodology.

Reply 1-0: We greatly appreciate the reviewer's kind comments on the significance of our work!

Changes made: Following the reviewer's suggestions, we have improved the descriptions of our methodology and addressed the reproducibility issue by making the data and custom codes/pseudo-codes available. Details are provided below.

Comment 1-1: ** data and source code availability (and statement?) I strongly encourage you to publish full supporting datasets and software along with this manuscript. This is the norm in the materials data science community, and is critical for the transparency and reproducibility of applied data science work, and can significantly increase your impact in the materials data science community by more effectively enabling others to adopt and extend your methodology.*

Reply 1-1: We fully agree on the importance of making data and code available. We have deposited the scanning electron diffraction datasets and the templates at University of Illinois Research Data Bank. The DigitalMicrograph (DM) scripts for cepstral analysis is also included. The template analysis part is based on methods described in a previous publication (Ref. 30, main text). These are available on individual basis upon reasonable request.

Changes made: We have added following data availability statement: "**Data availability:** The hyperspectral electron diffraction datasets, electron-diffuse-scattering templates, cepstral and Patterson function analysis code are available at the link, https://doi.org/10.13012/B2IDB-4432073_V1 (available starting Oct. 1, 2022, the materials can be downloaded temporarily from <https://uofi.box.com/s/5fbkes5b4yr74kotqs4a0lw94gwca15k>). The strain analysis tool, imtoolbox, can be downloaded from github at <https://github.com/flysteven/imToolBox>. Other data and code that support the analysis within this paper are available from the corresponding author upon reasonable request."

Comment 1-2: *I think revising Figure 1 somewhat to more clearly illustrate *how* the results are obtained would be very helpful for this manuscript, since some basic understanding of how the template-matching algorithm works will be useful for the context of this manuscript, which really does highlight the novel methodology. The current figure 1 is hard to understand at this level without fully reading and understanding the supplementary materials.*

I think this could be clarified by revising a few of the panels and improving the figure caption. Alternatively, step 4 could be its own figure. I would suggest: The panel titled "Nanocluster Detection": show all detections using the same color since this step comes before template identification and assignment. The panel titled "Nanocluster Diffuse Patterns" could be more clearly named "Identify pattern templates" or similar.

Finally, the panel at bottom right could show a schematic image with detections of each class, annotated with the colors corresponding to the template patterns shown in the top right panel. I would recommend using the image at bottom right with 10 yellow nanocluster detections for both the detection and classification/assignment panels, changing the colors of the detections in the assignment panel.

I think some revisions along these lines would go a long way towards clarifying the method.

Reply 1-3: We thank the reviewer for the detailed suggestions on improving our Figure 1. Based on the feedback, we have made following changes to the Figure. The figure is now divided into four panels for (1) Data acquisition, (2) Nanocluster detection, (3) Pattern grouping, and (4) Template identification and refinement.

The revised figure is shown below. Panel 1 is unchanged from the previous figure. Panels 2 illustrates the nanocluster-detection process from the recorded diffraction patterns (DPs), using the cepstral transform and cepstral annular dark-field (ADF) imaging. Two key steps are shown in Panel 3. The first is to remove the Bragg peaks from the recorded diffraction patterns, and the second is to use the cepstral ADF to identify nanoclusters and obtain the cluster DPs by averaging and grouping these patterns. The last panel shows the last step of the template identification and refinement, in which the diffuse DP for the identified type of short-range ordering is measured.

Fig. 1. Data mining of diffraction patterns (DPs) for CSRO detection in CrCoNi. Four steps (a to d) are involved. (a) Data acquisition, energy-filtered diffraction patterns (DPs) in a scan are recorded using an ~ 1 nm sized electron probe, with a pixelated detector. (b) The DPs are transformed into difference cepstra, which are then used to from the cepstral annular dark-field (ADF) image for nanocluster (NC) detection. (c) Bragg reflections are removed, using the log method, and diffuse DPs belonging to NCs are averaged and grouped together. (d) Diffuse DP templates are identified and used to detect the CSRO-strengthened NCs by the correlation method. The diffuse DPs from the same CSRO-strengthened NCs are then averaged, and the process is iterated to yield the refined templates. N is the number of collected DPs

(10^4), M is the number of distinct cluster diffuse DPs (10^1), and n ($n^2 = N$) is the number of scan points. ADF stands for annular dark field imaging, using the cepstral signals here.

Changes made: Figure 1 is replaced with the new one together with the revised caption.

Comment 1-2: *The data mining approach is not discussed in detail in the manuscript, and is not fully specified in the supplementary materials. Since the materials science results are enabled by this novel methodology, it is important to discuss in detail the specific details of each component of the analysis pipeline.*

*** peak detection algorithm document which algorithm is used, and any important hyperparameter settings, as well as a brief discussion of the criteria / data used to select.*

*** template generation and matching algorithm*

##+BEGIN_QUOTE

First, three diffraction templates belonging to CSRO are created from the cluster averaged log difference DPs

##+END_QUOTE

how does this work? manual identification of templates from a list of nanocluster average log-difference diffraction patterns? Random selection? Initialization from k-means clustering? Lots of interesting possibilities

The number of templates to include seems to be an important hyperparameter. How was the value of 3 selected and validate? Strong prior information based on a physical hypothesis is an acceptable justification, but warrants an explicit argument.

Next, you template-match individual diffraction patterns with cross-correlation against the template patterns. What is the value of the cross-correlation threshold, and how was it selected and validated?

Patterns attens selected through this procedure are averaged to refine the templates, in an iterative procedure? It's unclear from the manuscript whether templates are refine once or multiple times (until what criterion is met?)

The connection to k-means clustering is not clear, and I think it would be worth discussing in a bit more detail which aspects are similar. Certainly there are similarities in the cluster/template assignment step, and the iterative updating/refinement of the templates. Important differences to discuss seem to be 1. initialization strategy and 2. your method does not assign every instance to a template, in contrast to k-means which does.

*** oxide filtering. It's not explicitly discussed in the methodology -- how are oxide nanoclusters filtered out before performing template generation and matching? Are nanocluster detections manually flagged by inspecting diffraction patterns? That is implied by the discussion of indexing NiO reflections. Is there an automated procedure used to match and remove oxide nanoclusters before the template generation and matching process?*

Reply 1-2: We thank the reviewer for the detailed questions here. To address them, we have revised our

Supplement Note 1 to provide further details about the methods employed in our analysis. The revised note includes following new sections that are specifically related to the reviewer's questions:

- C. Identification of nanoclusters
- D. Nanocluster diffraction patterns and grouping
- E. Identification of diffuse diffraction patterns and surface oxides
- F. Correlation imaging and the measurement of electron diffuse scattering

Section C describes the procedures that we used for the nanocluster detection based on automated peak finding and followed by interactive adjustments. The latter was made mostly to the nanocluster size. The parameters for peak finding are also given in this section.

Section D expands the previous description provided in Ref. 30 of the main text to provide additional discussions about the template-matching method and diffraction pattern grouping, and how these methods work. Comparison with the k -means method is also made in this section. On the reviewer's question, indeed, random selection is used to initialize the clustering process. We use a correlation threshold to define the starting clusters, and the number of clusters is further refined during iterations (2 to 3 times, typically).

Section E describes the identification of DPs containing surface oxides and separate them from these with chemical short-range ordering. This process was performed manually by examining the DPs obtained after grouping. The identification of surface oxides was helped, using selected area diffraction, which yields the polycrystal diffraction rings. Employing this knowledge, we then mark the DPs with extra diffraction spots sitting on these rings as belonging to surface oxides. CSRO gives single-crystal diffuse patterns with broad peaks at $(111)/2$, (001) , and (110) positions for the $[1-10]$ zone axis. These features were used for the identification of DPs with CSRO. In selecting the starting CSRO templates, we looked for the diffusescattering patterns with the above characteristics and as symmetric as possible.

The template-refinement procedure is described in Section F. This procedure is based on the template-marching method described in Section D and peak finding method in Section C.

The above steps involved a mixture of automation and manual adjustments. Full automation of diffuse-scattering analysis likely will be challenging since as diffuse scattering is generally weak, but we agree that there are many future development possibilities to the data mining of 4D-STEM datasets.

Changes made: Following the reviewer's comments, we have added new sections (C, D, E, and F) to Suppl. Note 1 with the details and parameters about our data-mining procedure.

Reviewer #2 (Remarks to the Author):

Comment 2-0: *This manuscript reports two types of nanocluster short-range-orders (SRO) in an important single phase FCC medium entropy alloy CrCoNi. The unique properties of this alloy has been proposed to originate from SROs but the type of ordering structures has not been identified previously. The work is similar to the Nature work from Minor's group in 2020. Both works tackle the SRO issue with state of the art TEM tools and then show that SRO can be modified by processing techniques, and can affect mechanical behaviors. The conditions of samples are also exactly the same. Despite the fancy new techniques, the manuscript is not very well-organized. In particular, key information regarding the nanoclusters are confusing and self-conflicting in the present manuscript. The work may have new*

implications to the SRO in CrCoNi, but the authors must address the following issues before this work can be further considered for publication.

Reply 2-0: We thank reviewer for insightful comments and the time and effort in providing valuable feedbacks. In response, we have addressed the inconsistencies in our previous draft that the reviewer have pointed out and improved our manuscript.

Changes made: In response to the reviewer's comments, we have made corrections and modifications to improve our manuscript. We have also added an additional figure and related discussions based on the reviewer's suggestion. These changes are detailed in our replies below.

Comment 2-1: *The authors claim that whether the existence of SRO has not been confirmed in previous literature. Is it really so? Even in Minor's work? Or is the existence of SRO already confirmed but the structure remains unknown? Please clearly explain.*

Reply 2-1: We thank the reviewer for the questions. What we meant is that the existence of L1₂ chemical short-range ordering (CSRO) has not been confirmed by diffraction before. The observation reported by Zhang and Minor (*Nature* 581, 283-287, 2020) with the extra diffuse scattering along (1-11) and (-111) directions belong to the L1₁-type CSRO, according to our study. *The L1₂-type CSRO, characterized by diffuse scatterings at the (001) and (1-10) positions, was not observed by Zhang and Minor.*

Another major difference between our work and Zhang and Minor's work is that *they reported the presence of CSRO only in the heat-treated and "slow cooled" sample, but not in the water-quenched sample. We observed CSRO in both water-quenched and heat-treated samples.*

Changes made: Based on the reviewer's suggestion, we have modified our previous sentence "... However, it is far from being clear whether these observations support the theory-predicted CSRO^{12,21,23}." On Page 3 to following: "...However, the experimental short-range-order (SRO) parameters were not determined, and it is far from being clear whether these observations support the theory-predicted CSRO^{12,21,23}"

We added further clarifications in following places:

- (1) Abstract is modified with "Two samples, one homogenized and one heat-treated, are observed. In both samples, results reveal two types of short-range-orders inside nanoclusters that minimize the Cr–Cr nearest neighbors (L1₂) or segregate Cr on alternating close-packed planes (L1₁). The L1₁ is predominant in the homogenized sample, while the L1₂ formation is promoted by heat-treatment, which is accompanied by a large change in dislocation-slip mechanisms."
- (2) Figure 2 caption, we added "*The strong diffuse scattering is found in nanoclusters and their dominance changes from being Types A and B in Sample WQ to being Type C in Sample HT.*"
- (3) Figure 3 caption, *Chemical short-range ordering is observed in both samples with the L1₂ type dominantly found in Sample HT.*
- (4) Discussion, page 13, we added "The presence of CSRO in the homogenized sample was not detected by electron imaging, previously¹⁹."

Comment 2-2. *In general, one would expect the alloys to behave more like a random solid solution at higher temperature. This is exactly what Minor's work observed. Here, however, the NCs were larger, and the local strain was more uneven in the WQ sample, which suggests the opposite. The authors should explain and discuss this in detail. They should also discuss how the degree of SRO change with temperature in the alloy.*

Reply 2-2: We thank the reviewer for raising an important question here. Regarding *Zhang and Minor's work, they relied on selected area electron diffraction and does not have the high spatial resolution and high signal and noise ratio as in our scanning electron nanodiffraction technique.*

We also note that CSRO in the homogenized and water-quenched sample has also been reported by Inoue et al. (Physical Review Materials 5, 085007, 2021), which supports our observation, but contradicts the imaging result of Zhang and Minor.

On why the high-temperature phase is not retained during rapid quenching, we note that previous studies have shown that this highly depends on the SRO kinetics, which is very much materials dependent. For example, in $\text{Ni}_{0.765}\text{Fe}_{0.235}$, Lefebvre et al. (MRS Online Proceedings Library 21, 369-373, 1983) reported that the state of order characteristic at temperatures above T_c (ordering temperature) is retained upon quenching to room temperature, while Lucke et al. (J. Phys. Chem. Solids 37, 979-987, 1976) shows that the SRO forms in Au-Ag and Au-Pd when the rate of quenching from the annealing temperature to the temperature of measurement is not high enough compared to the rate of diffusion. The rate of diffusion is greatly influenced by quenched-in vacancies, which can have a large impact upon the kinetic of SRO formation. For example, in CuPt, with the $L1_1$ -type CSRO, it has been established based on resistivity measurements that each change in the annealing temperature is followed by a rapid change in the degree of SRO towards its equilibrium value (Urban-Erbil, B. & Pfeiler, W. in Ordering and Disordering in Alloys (ed A. R. Yavari) 164-171, Springer Netherlands, 1992).

Changes made: Based on the reviewer's suggestion, we added "The result (**Fig. 5**) reveals both $L1_1$ - and $L1_2$ -type CSRO in regions of a few nm in sizes. This observation supports our diffraction analysis and also the recent discovery of $L1_2$ CSRO in the homogenized sample by Inoue et al., using the lattice-resolved APT⁴⁰. The observation of CSRO in the homogenized samples shows that CSRO forms rapidly during cooling, similar to Au-Ag and Au-Pd⁴¹. The rapid formation is possibly assisted by the quenched-in atomic vacancies, which has been shown to have a large effect on the CSRO formation kinetics⁴¹."

Comment 2-3. *Why were the NCs not observable by HR or HRSTEM in the present work? In particular, both AP and HRTEM/HRSTEM should be able to image at least the $L1_1$ NCs, which were 6 nm in size. Such images could provide direct evidence for the existence of NCs and could also provide useful information of the SRO structure.*

Reply 2-3: We thank the reviewer for the suggestion. On why the CSRO NCs were not **directly** observable on **as-recorded** atomic resolution HAADF-STEM images, there two reasons. One is the small Z difference among Cr, Co, and Ni, which results in a small difference in the HAADF-STEM atomic column intensities. The second is the noise in the recorded image intensities, which obscure the small intensity difference.

To address the reviewer's question here, we performed further HAADF-STEM imaging experiment at a lower magnification than the imaging analysis described in Suppl. Fig. S1 and analyzed the result using template matching. The details of our method are described in the new Suppl. Note 8. This method measures sensitively the degree to which an atomic column deviates from the average. Using this method, we can detect SRO in Sample WQ as the reviewer suggested. The result is described in the new Fig. 5 below and related discussions.

Fig. 5 Color-coded cross-correlation image shows the coexistence of two types of SRO in Sample WQ. The image is obtained by template matching of the spatially averaged atomic column with the recorded HAADF-STEM image, using the procedures described in *Suppl. Note 8*. The $L1_1$ and $L1_2$ SRO are detected in two boxed regions, as evidenced by their power spectra. The cross-correlation coefficient is normalized with the mean equal to 1.

Changes made: We added the new Fig. 5 to the main text and added the related discussions: “... Our as-recorded HAADF STEM images also show no obvious difference between two samples (*Suppl. Fig. S1*). To further investigate these issues, we performed the cross-correlation analysis on atomic-resolution high-angle annular dark-filed (HAADF) STEM images of Sample WQ using template matching (*Suppl. Note 8*). A spatially averaged single-atomic-column image is used as the template. The resulted cross-correlation coefficient image measures sensitively the degree of atomic-column deviations. The result (**Fig. 5**) reveals both $L1_1$ - and $L1_2$ -type CSRO in regions of a few nm in sizes. This observation supports our diffraction analysis and also the recent discovery of $L1_2$ CSRO in the homogenized sample by Inoue et al., using the lattice-resolved APT⁴⁰.”

Comment 2-4. Could you provide a rough idea of the thickness of the TEM foil?

Reply 2-4: We measured the sample thickness using the convergent-beam electron diffraction (CBED) method. The regions where SEND patterns were taken are 40 to 50 nm in thickness (*Fig. R1*). The HAADF-STEM image was recorded in the areas close to the SEND analyzed regions.

Fig. R1 Sample thickness determination of areas in Sample WQ and Sample HT, where SEND data was collected.

Changes made: We added following in the Methods section: “The DP datasets were collected in 100 x 100-pixel scans over sample areas of 100 x 100 nm² and 40 to 50 nm in thickness.”

Comment 2-5. *Information regarding the composition of the NCs were confusing. On the one hand, the PDF suggest Cr depletion in the NCs for both samples. The atomic structural models in Fig. 3 are also based on off-stoichiometric compositions – but towards the opposite direction (Cr-rich side). On the other hand, the L12 NCs were described as “segregate Cr on alternating close-packed planes” – which can be achieved without any composition change in the NCs. The authors should explain these discrepancies and describe the composition of the L11 and L12 NCs. Also, if there is Cr depletion in the NC, there should be corresponding Cr-rich regions in the alloy, was those observed?*

Comment 2-6. Please provide Cr-centered PDF.

Reply 2-5,6: We appreciate the reviewer’s helpful feedback here. First, on the CSRO model composition, the models presented in Fig. 3 are selected from the theoretical models of both stoichiometric and off stoichiometric, which reproduced the experimental diffraction features. These models do not share the same composition, but rather highlight the possible range of compositions of CSRO strengthened nanoclusters. We made correction to clarify this point.

About the possible composition of CSRO nanoclusters from the experimental side, we know followings

- 1) At the tens of nm length scale, the alloy composition is uniform according to STEM/EDX (Suppl. Fig. 13).
- 2) At few nm length scale, Ni-rich and Cr-poor NCs are detected, based on the partial radial distribution function (RDF) analysis of APT data (Suppl. Info. Note 7).

- 3) The elemental correlations are rather small, at $\sim 2\%$, which extend to few nm.
- 4) From the simulations, the L1₁-type CSRO strengthened NCs can be poor or rich in Cr or Ni or Co according to the first-principles structural models (Figs. 3a and b and Suppl. Fig. 11), while the L1₂ CSRO can occur in both stoichiometric and off-stoichiometric compositions (Suppl. Fig. 10). The models are 3x3x3 unit cells (about 1.5 nm) in size

For the reasons below, these results suggest that there are a range of compositions for the L1₁ and L1₂ CSRO, which is consistent with theory predictions.

On the Cr segregation, indeed the L1₁ CSRO with alternating close-packed planes can be achieved without any composition change in the NCs as the reviewer pointed out. The resulted Cr composition modulation is at the length scale of {111} d-spacing, which is too small to be detected by our APT analysis. The observed Cr-poor regions in Ni- or Co-rich nanoclusters are formed by chemical clustering, leading to fluctuations in local composition. The Cr enrichment happens in the matrix. Because the partial RDFs are normalized by the volume composition, which is dominated by the matrix (details are provided in Suppl. Note 7), the enrichment is not showing in the plotted partial RDFs.

We also note that the nanoclusters of several nm seen by diffraction are chemically heterogeneous as well, since there is no evidence of large chemical clusters in the APT data. In our diffraction analysis, areas with similar diffraction patterns are grouped together. These diffraction features can be produced by the elemental-rich or poor volumes of 3x3x3 unit cells, according to our diffraction modelling. Thus, the large nanoclusters detected by diffraction can contain a collection of smaller clusters, whose composition fluctuates.

The Cr-centered partial RDF is now provided in the revised Suppl. Fig. 4. We also provided further comments in the interpretation of partial RDFs in our reply to Reviewer 3 (Reply 3-5), who raised similar concerns.

Changes made: We made following changes to address the reviewer's comments:

- 1) We added the sentences of "*The models shown here are selected from an ensemble with the CSRO composition varies from being stoichiometric to off-stoichiometric, which can be poor or rich in Cr (for details, see Suppl. Note 3).*" to Fig. 3 caption
- 2) We changed Suppl. Fig. 4 from the partial RDF analysis in a 25-nm diameter volume to the whole APT volume. The Cr-centered partial RDF is now provided. The normalization of RDF is currently discussed in Suppl. Note 7 together with an updated Suppl. Fig. 4.
- 3) On Page 13, we added the discussions of "At the nm-length scale, the APT partial RDF analysis detects Ni-rich and Cr-poor NCs of a few nm in size (*Suppl. Note 7*). The volume-averaged elemental correlations are small, at $\sim 2\%$ or less above or below that of random solutions. The small correlations can be explained by the presence of both elemental-poor and rich NCs in a randomly mixed matrix. The alloy composition is largely uniform at the tens of nm-length scale (*Suppl. Fig. S13*). The NCs of several nm seen in Sample WQ by diffraction are areas with similar diffraction patterns. The same diffraction features can be produced by the nm-sized elemental-rich or poor volumes, according to our diffraction modelling. Thus, these large NCs likely contain smaller clusters, whose composition fluctuates."

Comment 2-7. *The authors claim that for L12 "segregate Cr on alternating close-packed planes", but it seems to me that the NC with Cr on alternating close-packed planes in Fig. 3d is L11. It there a mistake?*

Reply 2-7. The reviewer is right, we have corrected this mistake.

Comment 2-8. *Quality of the manuscript is not quite pleasing. For example, the authors say that “In Sample WQ, localized planar slips sometimes form dislocation multipoles (MPs) (Fig. 4a and Suppl. Note 8)”. But there is no Note 8 in the supplement. There are also quite a few typos, for example, “Pattern functions” in the caption of Fig. 3 is clearly wrong.*

Reply 2-8. We thank the reviewer for pointing out the mistakes that we made in our initial draft. We have addressed these issues in the revised submission.

Reviewer #3 (Remarks to the Author):

Comment 3-0. *The paper by Hsiao presents a novel data-mining approach of electron-nanodiffraction patterns, complemented by other experimental and simulation techniques, to investigate short-range ordering (SRO) in the Medium entropy CrCoNi alloy (MEA). The results reveal two types of SRO that can dramatically the dislocation-slip behavior and hence the mechanical response of the alloy. This work illustrates convincingly the relevance of this new approach to study SRO in MEA/HEAs alloys. The presence of two types of SRO is clearly demonstrated and the experimental observations support the theory-predicted CSRO. In addition, the writing is clear and the results are well presented. For these reasons, I am happy for the work to be published in Nature Communications, once the comments in the document attached are considered.*

Reply 3-0: We thank reviewer for the kind recommendation and comments above our work.

Changes made: We have revised our manuscript to address the reviewer’s comments. These changes are detailed below.

Comment 3-1. *The WQ sample was prepared in a very similar way to the WQ sample in the Zhang paper [2]. In principle a rapid quench to room temperature should suppress SRO formation. Do you have some insights to explain why a similar thermal treatment leads to the formation of SRO in one case and not in the other?*

Reply 3-1: We thank the reviewer for raising this great question. Indeed, our sample treatment conditions are very similar to the homogenization, followed by water quenching and heat-treatment processing conditions used by Zhang and coauthors (*Nature* **581**, 283-287, 2020). Zhang et al. concluded that there is no CSRO in the water-quenched sample using energy-filtered selected area electron diffraction (EF-SAED). This conclusion is opposite to our finding and the recent observation in the high-resolution APT experiment reported by Inoue et al (*Physical Review Materials* 5, 085007, 2021, which we found after submission).

On why we detected SRO in the water-quenched sample but not by Zhang et al., we suspect that it is due to how the diffraction techniques were applied, and the conditions that they were applied. Zhang used EF-SAED to observe diffuse streaks, while we used EF-SEND. EF-SEND has the advantage and sensitivity of a large diffraction pattern dataset. To address the reviewer’s question, we have repeated the EF-SAED experiment. We found that the same diffuse streak observed by Zhang et al. in the water-quenched sample in thin regions (Fig. R1). The diffuse streak is more visible in thin samples than in thick samples (Fig. R2). At the sample thickness of 70 nm, the diffuse features are largely suppressed. Thus, in comparing

diffuse scattering of different samples using EF-SAED, it is essential that the sample is thin and similar in thickness. Unfortunately, the sample thickness information was not given in Zhang's work, so we could not verify whether the experimental conditions also played a role in their conclusions.

Fig. R2, Measurement of diffuse streaks in CrCoNi in samples of different thickness using energy-filtered selected area electron diffraction. The plots below were made along the lines marked in the diffraction patterns.

The $L1_2$ CSRO in the water-quenched sample is also evidenced by Inoue et al (Physical Review Materials 5, 085007, 2021) using the lattice-resolved APT. Interestingly, Inoue et al. also did not detect CSRO by SAED.

We agree that the suppression of SRO formation during a rapid quench is generally expected, but exceptions have been reported for binary alloys. In our reply to Reviewer #2, who raised a similar question, we pointed out that the SRO forms when the rate of quenching from the annealing temperature to the temperature of measurement is not high enough, compared to the rate of diffusion. In CuPt, with the $L1_1$ type CSRO, it has been reported that each change in annealing temperature is followed by a rapid change in the degree of SRO towards its equilibrium value [Urban-Erbil, B. & Pfeiler, W. in *Ordering and Disordering in Alloys* (ed A. R. Yavari) 164-171, Springer Netherlands, 1992]. This trend was attributed to the influence of quenched-in vacancies upon diffuse rate (Lucke et al., *J. Phys. Chem. Solids* 37, 979-987, 1976).

In addressing the second reviewer's question, we performed further HAADF-STEM imaging experiment at a lower magnification than the imaging analysis described in Suppl. Fig. S1 and analyzed the result using template matching. This method measures sensitively the degree to which an atomic column deviates from the average. Using this method, we can detect SRO in Sample WQ as well. The further supports our diffraction analysis.

Changes made: We added following discussions and new Fig. 5 to further support our discovery. "The presence of CSRO in the homogenized sample was not detected by electron imaging, previously¹⁹. Our as-recorded HAADF STEM images also show no obvious difference between two samples (Suppl. Fig. S1). To

further investigate these issues, we performed the cross-correlation analysis on atomic-resolution high-angle annular dark-field (HAADF) STEM images of Sample WQ using template matching (*Suppl. Note 8*). A spatially averaged single-atomic-column image is used as the template. The resulted cross-correlation coefficient image measures sensitively the degree of atomic-column deviations. The result (**Fig. 5**) reveals both L1₁- and L1₂-type CSRO in regions of a few nm in sizes. This observation supports our diffraction analysis and also the recent discovery of L1₂ CSRO in the homogenized sample by Inoue et al., using the lattice-resolved APT⁴⁰. The observation of CSRO in the homogenized samples shows that CSRO forms rapidly during cooling, similar to Au-Ag and Au-Pd⁴¹. The rapid formation is possibly assisted by the quenched-in atomic vacancies, which has been shown to have a large effect on the CSRO formation kinetics⁴¹.”

We also added the new references by Inoue et al. (Ref. 40) and Lücke et al. (Ref. 41).

Comment 3-2. *Interestingly the two samples have a very similar mechanical response. However, it is also shown that planar slip is much more prevalent in Sample HT. Shouldn't this affect the mechanical response (in particular the work hardening behaviour) as it was observed by Zhang[2]?*

Reply 3-2: We thank the reviewer for another great question that is at the center of debate about effects of CSRO on mechanical properties of HEAs. For CrCoNi, based on prior works, we expected that the yield strength would be either (1) similar, as observed by Inoue *et al.*'s¹ and Yin *et al.*'s² or (2) with a small difference, as observed by Zhang *et al.*'s (*Nature* **581**, 283-287, 2020). Our results agree with case (1).

On why Zhang et al. observed a small difference, one possible explanation is the small size of their tensile samples, which is 5.1 × 0.8 × 1.6 mm. The CrCoNi after homogenization has large-sized grains (~ 800 microns). Thus, only a small number of grains are involved in the tensile samples used by Zhang et al.. To reduce the grain-size effect, we used a larger sample size with a gauge length section of 25.4 × 2 × 3 mm.

We suspect that the similar yield strength is because the L1₁ and L1₂ CSRO were observed in both WQ and HT samples. The chemical heterogeneity in the two samples is pretty similar according to our APT analysis.

On how CSRO impacts on work hardening, it is a difficult question that requires further study of plastically deformed samples. The results reported here are for slightly deformed samples. As the sample deforms further, other mechanisms could be activated also. Our preliminary study of heavily deformed CrCoNi samples suggest the formation of stacking faults as a possible mechanism (not reported here), but its interplay with dislocation slip and CSRO is not clear yet.

Changes made: On Page 13, we added following “The yield strengths of the two differently prepared samples are similar at 227 and 224 MPa for Sample WQ and Sample HT, respectively (*Suppl. Fig. S1*). The measured yield strengths are in between the values reported by Zhang *et al.*¹⁹ (205 and 255 MPa, respectively, for the homogenized and heat-treated samples). Inoue et al.⁴⁰ also reported the similar mechanical properties in their differently-treated samples. We attribute the similar yield strength of our samples to the presence of CSRO in both.”

Comment 3-3. *Even if the correlation is not perfect, it appears that the NCs rich region tend to experience a tensile strain, while the regions depleted of NCs experience a compressive strain.*

This is especially visible for the WQ sample (Fig2. b-c). Is this expected? Can the authors comment on this?

Reply 3-3: We agree. There is a correlation between the NC distribution and strain, although the amount of strain variations is small, less than 0.5%. The DFT calculations reported by Walsh et al. using 3 x 3 supercells show a range of tetragonal-like lattice distortions from a fraction of percent to about one percent (Walsh, F., Asta, M. & Ritchie, R. O., *PNAS* **118**, e2020540118, 2021). The strain correlation and the magnitude of strain variations are consistent with these calculations.

Changes made: We added following discussion on page 7 “The observed strain correlates with the distribution of CSRO-strengthened NCs, which suggests that these NCs are likely lattice distorted. This observation is consistent with the first-principles predictions of tetragonal-like lattice distortions in the CSRO models²³”.

Comment 3-4: *What is the origin of the splitting of the L1₂ diffuse peaks?*

Reply 3-4: Early work suggested that the diffuse scattering is related to the atomic-interaction potential (Cowley, J. M. *Physical Review* 77, 669-675, 1950 and Clapp, P. C. & Moss, S. C. *Physical Review* 142, 418-427, 1966). Krivoglaz suggested that the diffuse scattering reflects the form of the Fermi surface in the so-called the Fermi-surface-imaging theory (Krivoglaz, M. A. *Diffuse Scattering of X-Rays and Neutrons by Fluctuations*. Springer-Verlag, 1996). Oshima and Watanabe studied the short-range-order diffuse scattering from disordered Cu-Pd and Cu-Pt alloys systematically for different compositions (*Acta Cryst.* A29, 520-526, 1973). They observed two-fold and four-fold splittings at (100), (110), and equivalent positions in the composition ranges from about 13 to 60 at. % Pd in the Cu-Pd system and up to about 45 at. % Pt in the Cu-Pt system. The separation of the split maxima increases with the Pd or Pt content. They interpreted these results based on the Krivoglaz’s Fermi-surface-imaging theory. B. L. Gyorffy and G. M. Stocks showed that the observed concentration-dependent peaks in X-ray and electron diffuse scattering intensities are due to parallel sheets of flat Fermi surfaces (*Physical Review Letters* 50, 374-377, 1983). Their work was based on a first-principles electronic model for the forces driving clustering and short-range order. This explanation was further supported by the first-principles calculations.

The splitting that we observed in CrCoNi for L1₂ CSRO is consistent with what Oshima and Watanabe observed in Cu-Pd. This trend suggests that in addition to magnetic interactions, the Fermi surface instability also contributes to ordering.

Changes made: We added following discussion on Pages 12-13. “The splitting in the diffuse peaks from L1₂ CSRO is consistent with what Oshima and Watanabe observed in Cu-Pd, which is related to the Fermi surface (FS) shape, through the so-called FS nesting mechanism. Its observation suggests that in addition to magnetic interactions, the Fermi surface instability also contributes to CSRO.”

Comment 3-5: *I am a bit confused by the interpretation of the APT data. Previous work [1] clearly show that SRO involves a large decrease (~40%) in the number of Cr-Cr pairs together with a significant increase in Ni-Cr and Cr-Co pairs with respect to random distribution. The number of Ni-Ni pairs also decreases while the number of Co-Co pairs is not affected. Based on this I am not sure I understand why the Ni-Cr and Cr-Co correlations are so low and the Ni-Ni correlations so high? Can the authors clarify this (in case I misunderstood)?*

Reply 3-5: Thank you for pointing out this important point. The authors of Tamm et al.'s work used Monte Carlo methods coupled with density functional theory (DFT). The table below shows all the calculated SRO parameters:

Pair	500 K	800 K	1200 K
Ni-Ni	0.12	0.03	0.07
Ni-Cr	-0.27	-0.15	-0.12
Ni-Co	0.15	0.13	0.05
Cr-Cr	0.42	0.28	0.14
Cr-Co	-0.16	-0.12	-0.03
Co-Co	0.01	0.00	-0.03

Table R1. The calculated short-range order parameter of the CrCoNi ternary FCC system reported by Tamm et al. (Tamm et al., *Acta Materialia* 99, 307-312, 2015).

From our diffraction study, we obtained α_{Cr-Co} or α_{Cr-Ni} value at - 0.18 for the $L1_2$ CSRO, which is consistent with above calculations.

Regarding the difference between simulations and the APT data, our APT results do not resolve the crystal lattice (Suppl. Fig. S4). Rather, we are looking at compositional fluctuations averaged over 2 to 3 shells around each detected atom, while simulations investigate these first few shells around each atom. To be able to compare APT results with simulations, high-resolution APT is needed. This experiment has been done by Inoue et al. (*Physical Review Materials* 5, 085007, 2021) through the APT analysis of [100]- and [111]-oriented sharp needles in small volumes. Their one-dimensional spatial-distribution maps do show that the elemental density along the [001] direction is modulated with Cr-rich {100} atomic layers and (Ni + Co)-rich {100} layers aligned in the FCC structure, which is consistent with $L1_2$ CSRO.

We examine chemical fluctuations at several nm length scale in larger volumes than the study of Inoue et al. (*Physical Review Materials* 5, 085007, 2021). We used the partial radial distribution function (RDF) analysis that examines three-dimensional correlations. The analysis resolution is about 0.5 nm.

There are also several considerations in interpreting the RDF results. First, APT only detects part of atoms. Both the number of atoms detected, and their positions depend on the field evaporation process (Miller, M. & Forbes, R. *Atom-probe tomography: the local electrode atom probe*. Springer. 2014). For example, the z position is determined more precisely than the x and y positions due to the difference between distinguishing the sequence of electrical field-evaporated ions, and the trajectory-based determination for which aberrations associated with the electrical field-evaporation are unavoidable. Another issue needs to be considered is the different field-evaporation strengths of elements in multicomponent alloys. According to Hatzoglou et al. (<https://arxiv.org/ftp/arxiv/papers/2008/2008.08801.pdf>), Cr has a lower evaporation voltage than Ni. The atom maps exhibiting aberrations in the atom positions can result from the nonuniform field evaporation. Additionally, field evaporation can be impacted by the presence of vacancies or interstitials. A previous APT study of irradiated NiCoCr alloys found Ni and Co enrichment and Cr depletion around He bubbles, which was attributed to diffusion mechanisms, e.g., Ni and Co sites facilitate interstitial diffusion, while Cr sites facilitate vacancy diffusion (Wang, X. et al. *Nat. Commun.* 11,

1022, 2020). For these reasons, the Ni-Co RDF is different from the Co-Ni RDF and etc.. These RDFs would be the same if all atoms were detected.

In interpreting the calculated partial RDFs, another factor to consider is the density normalization. The details about this factor are now included in Suppl. Notes 7. Our previous calculation analyzed a 25 x 25 x 25 nm³ cube within the APT volume, while the composition used for normalization was taken from the entire APT volume, which introduced small differences in the composition at large distances. In this revised version, we calculated the partial RDFs for the entire APT volume to address the above issue. We have also included the Cr centered partial RDFs, as shown below.

Fig. S4. APT partial radial distribution functions (RDFs) of the heat-treated and water-quenched CoCrNi samples. (A, B, and C), and (E, F, and G) calculated partial RDFs with Co, Ni, and Cr as the center atom, respectively, as determined from the APT data, for heat-treated and water-quenched CoCrNi samples. Note that the partial RDFs were normalized by the averaged composition for the APT analyzed volume. (D) and (H) APT atom maps of the constituent elements (Co, Cr, and Ni) for heat-treated and water-quenched CoCrNi samples, respectively.

The calculated partial RDFs are averaged over a volume that can be considered as being made of a complex mixture of a matrix of a random solution and elemental rich or poor NCs. This complexity makes the interpretation of partial RDFs difficult. Nonetheless, the measurements show the strongest RDF signals in the Ni-Ni, Co-Cr, and Ni-Cr correlations and their deviation from the random solution. The net positive Ni-Ni RDF, which is stronger than the Co-Co and Cr-Cr RDFs in Fig. S4, can be taken as that there are more Ni-rich NCs than Ni-poor ones in a matrix that is slightly Ni poor. The negative Co-Cr and Ni-Cr correlations extending to ~1 nm for Co- and Ni-centered atoms indicate the presence of Cr-poor but Ni, Co, or (Ni, Co) rich small NCs. In comparison, the averaged Cr-Cr is close to the matrix composition. Its difference with the Ni-Ni RDF, especially, suggest an equal number of Cr-rich and poor regions with the caveat that the result may also reflect the different role of Cr and Ni in atomic diffusion.

Changes made: We included additional discussions on the calculation of the APT RDF, interpretation of RDF curves, and comparison with Inoue et al.'s work in the expanded Suppl. Note 7.

The following paragraph is added on the interpretation of APT partial RDFs in main text:

“At the sub-nm scale, CSRO minimizes Cr–Cr nearest neighbors ($L1_2$) or segregate Cr on alternating close-packed planes ($L1_1$). While these two types of CSRO can be formed without altering the chemical composition, diffraction modelling suggests that the CSRO-strengthened NCs are likely off-stoichiometry. For example, the diffuse peaks at $\{111\}/2$ positions (**Fig. 2a**), which characterize the $L1_1$ -type ordering, are reproduced by diffraction simulations in the off-stoichiometric, quasi-random, models constructed by Walsh et al.²³. The configurations in these models are either Cr or Co/Ni poor, and their formation is likely associated with the elemental diffusion driven by magnetic interactions.

At the nm-length scale, the APT partial RDF analysis detects Ni-rich and Cr-poor NCs of a few nm in size (*Suppl. Note 7*). The volume-averaged elemental correlations are small, at $\sim 2\%$ or less above or below that of random solutions. The small correlations can be explained by the presence of both elemental-poor and rich NCs in a randomly mixed matrix. The alloy composition is largely uniform at the tens of nm-length scale (*Suppl. Fig. S13*). The NCs of several nm seen in Sample WQ by diffraction are areas with similar diffraction patterns. The same diffraction features can be produced by the nm-sized elemental-rich or poor volumes, according to our diffraction modelling results. Thus, these large NCs likely contain smaller clusters, whose composition fluctuates.”

Comment 3-6: *There are few typos, and a missing table:*

Main:

Page 11 – l 236 : “form dislocation multipoles (MPs) (Fig. 4a and Suppl. Note 8)” → **there is no supplementary note 8**

Supporting information

Page 5 – l 81: “ A majority of NCs seen in Fig. 6C are surface oxides”, → **should be Fig. S6C**

Page 17 – l 266-267 : “or (A) Area1 and (B) Area2 in Suppl. Fig. 2C” → **should be Fig. S6C**

Reply 3-6: We thank the reviewer for your careful reading of our manuscript and pointing out these mistakes. These errors have been corrected in the revised manuscript.

REVIEWERS' COMMENTS

Reviewer #1 (Remarks to the Author):

I found the revisions very thorough, and am satisfied that everything has been appropriately addressed.

Reviewer #2 (Remarks to the Author):

This reviewer is in general pleased with the responses of the authors. Particularly the newly added Fig. 5, which provides a more direct evidence for the NCs. The manuscript is now recommended to be published in Nature Communications.

Reviewer #3 (Remarks to the Author):

The authors addressed my questions and comments convincingly. At this stage I would recommend this work for publication in Nature Communications. I just have a couple of comments / remarks.

1) Figure 1 was modified following the suggestions of reviewer #1. These modifications were mostly beneficial, however I would say this figure might still be difficult to understand without a careful reading of the supplementary materials.

The arrows connecting the Cepstral ADF and the cluster diffuse DPs are a bit difficult to see. In my opinion, the previous version of the figure, where several nanoclusters were shown on cepstral ADF together with their corresponding cluster diffuse DPs, was easier to understand.

Similarly, I would have kept the yellow circles in the correlation map showing the nanoclusters with the same CSRO.

2) There are still a few typos in the supplementary materials:

a) Note 1 - Section E should be section F

b) Page 6 - l 126 : "A majority of NCs seen in Fig. 6C are surface oxides" -> should be Fig. S6C